# Clays, Limestone and Biochar Affect the Bioavailability and Geochemical Fractions of Cadmium and Zinc from Zn-Smelter Polluted Soils

**Altaf Hussain Lahori [1,2,*]**, **Monika Mierzwa-Hersztek [3,4]**, **Erdona Demiraj [5]**, **Rachida Idir [6]**, **Thi Tuyet Xuan Bui [7]**, **Dinh Duy Vu [8]**, **Amanullah Channa [9]**, **Naeem Akhtar Samoon [2]** and **Zengqiang Zhang [1,*]**

[1] College of Natural Resources and Environment, Northwest A&F University, Yangling 712100, China
[2] Department of Environmental Sciences, Sindh Madressatul Islam University, Karachi 74000, Pakistan; naeemsamo@smiu.edu.pk
[3] Department of Agricultural and Environmental Chemistry, University of Agriculture in Krakow, al. Mickiewicza 21, 31-120 Krakow, Poland; monika6_mierzwa@wp.pl
[4] Department of Mineralogy, Petrography and Geochemistry, al. Faculty of Geology, Geophysics and Environmental Protection, AGH University of Science and Technology, Mickiewicza 30, 30-059 Krakow, Poland
[5] Department of Agro-Environment and Ecology, Faculty of Agriculture and Environment, Agriculture University of Tirana, Tirana 1029, Albania; edemiraj@ubt.edu.al
[6] Cerema, Project-Team DIM, 120 Rue de Paris, BP 216 Sourdun, 77487 Provins CEDEX, France; rachida.idir@cerema.fr
[7] Institute of Ecology and Biological Resource, Vietnam Academy of Science and Technology (VAST), No. 18 Hoang Quoc Viet, Caugiay, Hanoi 100000, Vietnam; tuyetxuansttv@gmail.com
[8] Vietnam–Russia Tropical Centre, 63 Nguyen Van Huyen, Nghia Do, CauGiay, Hanoi 100000, Vietnam; duydinhvu87@gmail.com
[9] Department of Economics, Abdul Haq Campus, Federal Urdu University of Arts, Science & Technology, M.A. Jinnah Road, Karachi 75300, Pakistan; dr.amanullah@fuuast.edu.pk
* Correspondence: ahlahori@yahoo.com (A.H.L.); zhangzq58@126.com (Z.Z.); Tel.: +86-136-0925-4113 (Z.Z.)

**Abstract:** Ca-bentonite (CB) alone and in a mixture with limestone (L), tobacco biochar (TB) and zeolite (Z) on the fixation, geochemical fractions and absorption of Cd and Zn by Chinese cabbage in smelter heavily polluted (S-HP) and smelter low polluted (S-LP) soils were investigated. The results showed that the CB + TB and CB + L + TB treatments significantly immobilized Cd up to 22.0% and 29.7%, respectively, and reduced uptake by Chinese cabbage shoot to 36.0% with CB + Z + L and 61.3% with CB + L in S-HP and S-LP soils compared with the control. The CB + Z + L + TB treatment mobilized Cd up to 4.4% and increased absorption in the shoot by 9.9% in S-HP soil. The greatest immobilization of Zn was 53.2% and 58.2% with the CB + Z + L + TB treatment, which reduced Zn uptake in the plant shoot by 10.0% with CB + L and 58.0% with CB + Z + L + TB in S-HP and S-LP soils. The CB + Z + TB and CB + TB treatments mobilized Zn up to 35.4% and 4.9%, respectively, in both soils. Furthermore, the uptake of Zn in plant shoot was observed by 59.0% and 7.9% with application of CB + Z and CB + TB treatments, respectively, in S-HP and S-LP soils. Overall, our results suggest that Ca-bentonite alone and in mixtures with different amendments can be used to reduce the phyto-extraction of Cd and Zn in Zn-smelter polluted soils.

**Keywords:** clay minerals; limestone; biochar; trace elements; bioavailability; alkaline degraded soils

## 1. Introduction

Global industrialization and anthropogenic activities, such as smelting, mining ores, modern agriculture practices and waste disposal methods, have rapidly contaminated soils with potentially toxic elements (PTEs) [1]. PTEs are important inorganic pollutants in soils, because they are potentially toxic for animals and humans if they enter the food chain [2]. Cadmium (Cd) is recognized as one of the extremely movable and possibly bio-available soil toxic substances, and its prospective role in plant metabolism and physiology has not been reported. Therefore, we understand that the bioavailability of all PTEs depends on the Cd accessibility in soil and plant genotypes [3]. In addition, zinc (Zn) is considered a pivotal micronutrient for plant growth and development, especially in paddy Zn-deficient soils, but its toxicity may have a negative impact on living organisms [4]. The poisonous nature of PTEs is contingent on numerous soil physicochemical conditions, viz. pH, temperature, ion solubility, TE forms and concentration [5].

Although the cleanup of PTE soils is very necessary, the implementation of most conventional reclamation approaches, viz. encapsulation, land filling, electrokinetics, surface capping, vitrification and soil washing/flushing, are economically nonfeasible on a large scale since these methods are environmentally disturbing and too expensive [6]. Amongst these technologies, in-door fixation of PTEs is thought to be the best opportunity and a robust technology for restoration of inorganic polluted sites [7]. The principal mechanism of PTE fixation in soil depends on (ad) sorption, pH, CEC, electrostatic attraction, temperature, redox-potential and precipitation [8]. It is vitally important to bring PTE infertile polluted soils under cultivation using feasible and cheap amendments. Nonetheless, effective low-cost practices are greatly needed to resolve this global problem [9]. The in-situ fixation of PTEs by organic or synthetic additives, e.g., clay additives [10,11], phosphate compounds [12], bio-solids [13], and alkaline material [14], is a promising option for cleanup of inorganic pollutants in contaminated sites because this method is cheap, feasible and environmentally friendly [15]. Subsequently, Chaves and Tito [16] stated that reactions of PTEs with clay additives are very imperative for influential metal compounds in the surroundings. Moreover, Yi [17] reviewed that clay additives have great potential to enhance soil pH, which may account for reduced chemical-extractable fractions and the bioavailability of metal-polluted soil and restrict their accumulation in plants.

In recent years, a significant number of studies have been performed using single agents or mixed/combined agents for reclamation of PTEs from artificial or historical contaminated soils. Indeed, several researchers concluded that alkaline amendment could be successfully used for restoration of acidic soils by increasing the soil pH [18,19]. Paz-Ferreiro [20] investigated the immobilization mechanism of PTE's between phytoremediators and biochar. Gasco [21] examined the combined effect of phytoextraction by *Brassica napus* and rabbit-made biochar as additive for the reclamation of a mine-polluted soil. Patel [22] studied the slow pyrolysis of biosolids in a bubbling fluidized bed reactor and compared with poor cost bed materials, viz. lime, biochar and activated char; furthermore, the results found that bed materials produced high surface area biochar between temperatures of 700 and 900 °C and decreased oxygenated, PACs, nitrogenated and aliphatic compounds in bio-oil. Recently, Álvarez [23] reported that poultry and rabbit manure-made biochar pyrolysis at 450 and 600 °C significantly reduced the Ni, Zn, Cd, Pb and Cr mobility in acidic soil as compared to alkaline old mine-polluted soil. Cardenas-Aguiar [24] assessed the potential of manure waste-made hydrochars on the remediation and uptake of Zn, Pb and As by *Brassica napus* in two mine-polluted soils.

Limited research work has been done to investigate the additive efficiency of mixtures of agents on the phytoextraction of PTEs in soils with elevated pH rather than acidic contaminated soil. To date, limited information is available on the efficacy of CB alone and mixed with limestone (L), zeolite (Z) and tobacco biochar (TB) on immobilization, geo-genic fractionation and uptake of Cd and Zn in alkaline Zn-smelter polluted soils. Therefore, it is hypothesized that the CB alone and in a mixture with L, TB and Z might be highly effective for the restoration of degraded alkaline polluted soils. Hence, the aims of current study were as follows: (1) to explore the efficacy of CB alone and in mixture with L, TB and Z for bio-availability, speciation, plant growth and absorption of Cd and Zn by Chinese

cabbage, (2) to elucidate their potential to remediate heavily and low polluted top soils for agricultural purposes, and (3) to probe the response of studied additives on soil health.

## 2. Materials and Methods

### 2.1. Collection of Contaminated Soils

Smelter heavily polluted (S-HP) and smelter low polluted (S-LP) soils were sourced from Feng county of Shaanxi Province, China. In fact, this area was extremely polluted with Cd and Zn because of the Zn-smelting plant. Therefore, to address this problem, S-HP soil was taken from (33°56′45″ N, 106°31′45″ E) nearby the Zn smelter, and S-LP soil was collected (33°57′42″ N, 106°31′46″ E) approximately 1.6 km away from Zn-smelter. The surface soil samples 0 to 20 cm deep were taken by using a shovel and then placed into polyethylene bags for pot experiment. The collected soils were manually crushed and sieved using a 2 mm mesh. Furthermore, selected basic physicochemical parameters of the two soils are indicated in Table 1.

**Table 1.** Selected physicochemical parameters of contaminated soils and additives.

| Parameters | S-HP | S-LP | CB | L | TB | Z |
|---|---|---|---|---|---|---|
| Textural class | Sandy loam | Sandy loam | - | - | - | - |
| EC (1:5) (dS m$^{-1}$) | 0.3 | 0.2 | 0.1 | 6.4 | 11.8 | 0.1 |
| pH (1:5) | 8.6 | 8.8 | 8.2 | 11.7 | 10.5 | 7.9 |
| Organic matter (%) | 0.17 | 0.6 | - | - | 63.0 | - |
| Cation exchange capacity (cmol kg$^{-1}$) | 21.4 | 29.2 | 51.0 | 0.1 | 49.1 | 6.1 |
| Dissolved organic carbon (mg kg$^{-1}$) | 20.8 | 26.1 | 70.1 | 8.4 | 21.4 | 79.1 |
| Total Cd (mg kg$^{-1}$) | 19.1 | 10.3 | 0.9 | 1.4 | 5.2 | 2.6 |
| Total Zn (mg kg$^{-1}$) | 519.2 | 234.3 | 65.2 | 6.2 | 29.4 | 17.2 |

Immobilizing Agents

Four common immobilizing additives were selected in this investigation: (a) Ca-bentonite (CB), (b) limestone (L), (c) tobacco biochar (TB) 500 °C, and (d) zeolite (Z). All studied amendments were purchased from different local traders of Yangling City, Shaanxi Province, China. The EC and pH were detected in a 1:5 H$_2$O ratio (w/v) for tobacco biochar and inorganic amendments using the USEPA9045D [25] Method and ASTMD1125 [26], respectively. The total PTE content in amendments were examined using the 3050B method USEPA [27], and assessed by AAS (Z-2000, Japan). The basic chemical properties of the additives are presented in Table 1.

### 2.2. Experimental Design

A pot study was performed at Northwest A&F University (NWAFU), Yangling (34°15′60″ N, 108°3′46″ E), Shaanxi province of China. Thus, all immobilizing agents were sieved <2 mm prior to application at (1% weight/weight). In this trial, Ca-bentonite (CB) alone and mixed with limestone (L), tobacco biochar (TB) and zeolite (Z) were used in 9 treatments: (T1) Soil (Control); T2 Ca-Bentonite (CB 1%); T3 Ca-Bentonite + Limestone (CB 1% + L 1%); T4 Ca-Bentonite + Tobacco biochar (CB 1% + TB 1%); T5 Ca-Bentonite + Zeolite (CB 1% + Z 1%); T6 Ca-Bentonite + Limestone + Tobacco biochar (CB 1% + L 1% + TB 1%); T7 Ca-Bentonite + Zeolite + Tobacco biochar (CB 1% + Z 1% + TB 1%); T8 Ca-Bentonite + Zeolite + Limestone (CB 1% + Z 1% + L 1%); and T9 Ca-Bentonite + Zeolite + Limestone + Tobacco biochar (CB 1% + Z1%+ L1% + TB 1%). The selected treatments and their doses were planned on the basis of earlier studies [18,28,29]. All additives were incorporated to 3.0 kg of soil in each plastic pot with a height of 16 cm and width of 23 cm; each treatment was replicated thrice in a complete randomized block with the exception of control pots. This design represented 54 pots in total (2 soils × 9 treatments × 3 replications × 1 crop). The soil and each immobilizing agent were assorted systematically and used in plastic pots. Approximately, 1 L of distilled water was

added in experimental pots to maintain a field capacity of ~65%. Experimental pots were incubated 40 days for chemical reaction under normal greenhouse conditions. After completion, the cultivation Chinese cabbage (*Brassica campestris* L.) hybrid F1 was chosen as a test plant in this investigation. Furthermore, pure seed was purchased from a local market of Yangling, Shaanxi China. Seed was soaked in de-ionized water for 6 hours. Then, 10 vigorous seeds after soaking were grown in each pot, and five plants were selected after two weeks. At the time of seed germination, 80% soil moisture was maintained, and the lost water was regularly adjusted every week for Chinese cabbage. The entire Chinese cabbage plants from both soils were carefully removed from all pots after six weeks of planting.

*2.3. Sample Analyses*

2.3.1. Soil Sample Analysis

The electrical conductivity of soil (EC) and pH levels were assessed at a ratio of (1:5 $H_2O$) using a mechanical pH and EC meter. Thus, OM was measured using the Walkley–Black titration method ISO 14235 [30]. Soil texture was assessed with a Master sizer 2000E (Malvern, UK) laser diffractometer [31]. The dissolved organic carbon (DOC) was detected in ultrapure water in a 1:10 soil-water ratio using a dry combustion protocol in an automated TOC analyzer (Shimadzu TOC-L). The CEC was detected following the USEPA Method 9080 [32]. The TE fractionations, such as F1 = exchangeable, F2 = Fe, Mg oxide bounded, F3 = organic matter bonded, and F4 = residual from contaminated soils, were detected using the BCR-sequential extraction method followed by [33]. Total (pseudo) PTE concentration in soils was detected followed by AAS (Z-2000, Japan) using ICP-OES.

2.3.2. Extraction of PTEs Using DTPA/TEA

The solubility of PTE's in soils was assessed in DTPA (diethylenetriaminepentaacetic acid extraction) using a modified method ISO 14870 [34]. The solution was analyzed for cadmium and zinc using a Polarized Zeeman Atomic Adsorption spectrometer (Z-5000).

2.3.3. Plant Analysis

Chinese cabbage crop was uprooted six weeks after being sown, and shoot and root biomass was separated from both soils, washed thoroughly with distilled water to avoid any contamination. Then, plant biomasses were carefully oven dried at 65 °C to a constant weight. The dried samples were ground with a stainless-steel grinder chamber and stored in polyethylene bags for analyses. Approximately 0.20 g of shoots and roots were weighted and digested using nitric acid ($HNO_3$–) and perchloric acid ($HClO_4$–). Total Cd and Zn contents in plant samples were tested using an Inductively Coupled Plasma mass spectrometer (ICP-MS) (ELAN DRC-e, Perkin Elmer SCIEX).

2.3.4. Chlorophyll Assessment

The chlorophyll content in plant leaves was detected before Chinese cabbage harvesting using SPAD-502, Osaka 590-8551, Japan (Soil Plant Analysis Development) [35]. Furthermore, aerial surface biomass of five plants chosen from each pot was assessed, and the total average is reported in this investigation.

*2.4. Quality Control and Statistical Analysis*

All examinations were performed in triplicates. Reagent blanks were used for accurate assessment of the analytical values. Soil GBW07457 (GSS-28) and plant GBW07603 (GSV-2 Chinese cabbage) certified reference materials from Chinese Academy of Geological Sciences were used for quality control. The recovery ratios of cadmium and zinc in soil ranged from 93–103% and 92–102%, respectively, and those in plants ranged from 94–104% and 96–106%, respectively. One-way analysis of variance (ANOVA) and least significant difference (LSD) test were employed to identify significant differences between individual treatments ($p < 0.05$) using SPSS version 22.0. All graphs were completed using

Origin 17.0 software. Redundancy analysis (RDA) between DTPA-extractable Cd and Zn, uptake in root and shoot biomass, plant dry biomass, chlorophyll content, and soil chemical index were also performed using CANOCO5.

## 3. Results and Discussion

### *3.1. Plant Growth*

The maximum shoot and root biomass of Chinese cabbage from (S-HP) soil was significantly increased by 19.9 and 5.8% with CB + TB (T4), whereas the CB + L + TB (T6) and CB + L (T3) treatments reduced the shoot and root biomass of Chinese cabbage by 70.0 and 58.1% compared with control (Figure 1a). Regarding (S-LP) soil, the greatest shoot and root biomass of Chinese cabbage was observed at 27.2 and 23.1%, respectively, with CB + TB (T4), while the CB + L (T3) treatment significantly reduced these levels by 60.6% and 60.0%, respectively, compared to control (Figure 1b). Our findings clearly showed that changes in plant dry weight might result from variations in soil pH and low mobility of nitrogen (N) due to the introduction of alkaline additives. In an earlier study, Cheng and Hseu [36] reported that zeolite and bentonite as soil additives did not support cabbage growth due to the elevated soil pH. In addition, Yi [37] observed an increase in *Brassica juncea* dry biomass with increasing dosages of bentonite and zeolite compared with control.

### 3.1.1. Chlorophyll Content

The chlorophyll amount in the Chinese cabbage shoot from S-HP soil was significantly increased from 35.0 to 39.5 (approximately 11.39%) with CB + Z (T5), but the CB + Z + L (T8) treatment significantly reduced the amount of chlorophyll from 35.0 to 33.3 (4.9%) compared with control (Figure 1c). The chlorophyll content in the Chinese cabbage shoot from S-LP soil was significantly increased from 37.7 to 42.4 (11.2%) with CB (T2), while the greatest reduction of chlorophyll content in plant shoot was observed from 37.7 to 32.9 (12.8%) with CB + Z + TB (T7) compared with control (Figure 1c). It is assumed that the changes in chlorophyll content might be due to variations in the soil chemical index after the addition of alkaline amendments. Our findings agreed with previous studies [35,38,39].

### 3.1.2. Soil Organic Matter

The organic matter (OM) concentration in S-HP increased from 0.14 to 0.26% with CB + Z + TB (T7), whereas CB + Z (T5) reduced it from 0.14 to 0.07% compared with the unamended treatment (Figure 1d). Compared with the control, the CB + L + TB (T6) treatment was highly effective for increasing OM from 0.58 to 0.84%, but CB (T2) significantly reduced the OM percentage from 0.58 to 0.13% in S-LP soil (Figure 1d). Our results clearly indicated that original smelter-contaminated soils were poor in OM, while the CB + Z + TB (T6), and CB + L + TB (T7) treatments were highly effective in increasing the OM concentration with the addition of TB in mixed treatments. Abdelhafez [40] revealed an increase in OM in smelter-polluted soil after the introduction of sugar cane bagasse and orange peel-derived BCs. Zhang [41] used rice straw-derived BC as an amendment that evidently increased OM from metal-contaminated soil. Additionally, Shuman [42] stated that the soil organic matter (SOM) demonstrated the potential to improve PTE toxicity to plants by re-distributing elements to less toxic conditions. Upon the normal application of OM into the soil, most nutrients were fixed [43,44]. Conversely, as the OM decays, the adsorbed PTEs are solubilized [45].

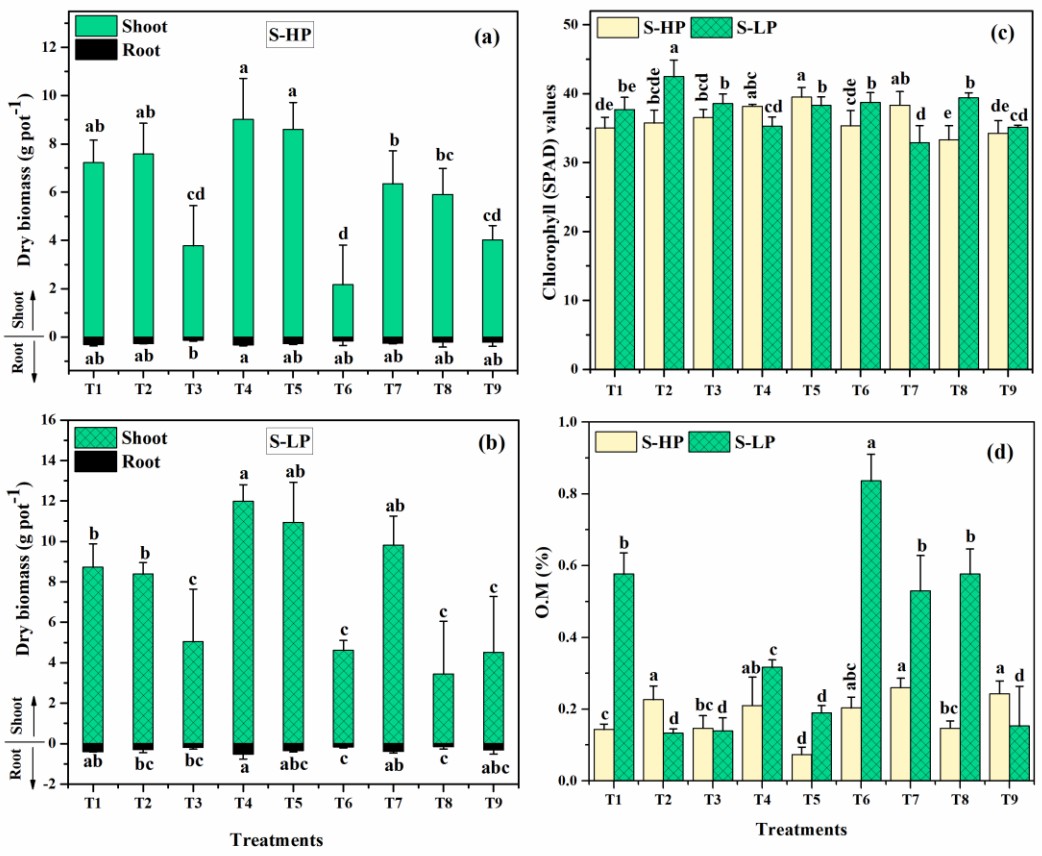

**Figure 1.** Impact of soil additives on (shoot and root) dry biomass yield of Chinese cabbage grown in Fengxian heavily contaminated (S-HP) soil (**a**), Fengxian low contaminated (S-LP) soil (**b**), Chlorophyll soil plant analysis development (SPAD) values (**c**), and organic matter (**d**). Values in a given column followed by the same letter are not significantly different ($p < 0.05$) using Tukey's LSD test.

### 3.1.3. Soil Electrical Conductivity

Compared with control, the EC level in S-HP soil was nonsignificantly increased from 0.30 to 0.40 dS m$^{-1}$ in CB + Z + TB (T7), while CB + Z + L (T8) reduced the EC level from 0.30 to 0.17 dS m$^{-1}$ (Figure 2a). The EC level in S-LP soil significantly increased from 0.38 to 0.40 dS m$^{-1}$ with CB + TB (T4), while the maximum reduction of EC was observed from 0.38 to 0.17 dS m$^{-1}$ in CB + L (T3) compared with control (Figure 2a). Mohamed [18] demonstrate that bamboo BC as an amendment can increase soil EC levels compared with control soil. Consequently, Meng [46] revealed that the co-pyrolysis of rice straw with swine manure-derived BCs at 400 °C evidently increased the EC level in Pb-Zn mining-polluted soil.

### 3.1.4. Soil pH

The pH values in S-HP soil significantly increased from 8.47 to 9.10 with CB + Z + L (T8), whereas CB (T2) reduced the pH values from 8.47 to 8.43 compared with unamended soil (Figure 2b). The potential role of amendments to S-LP soil significantly increased the pH level from 8.54 to 8.94 with CB + Z + L + TB (T9) compared with nonamended soil (Figure 2b). Overall, it is confirmed that the increasing soil pH might be due to co-application of alkaline amendments. Li [47] reported an increase in soil pH with the addition of zeolite. Likewise, Wu [48] reported that increased soil pH and OH$^-$ levels might possibly increase the precipitation of carbonates or hydroxides. Moreover, Boostani [49] reported that the addition natural zeolite and biochars in contaminated soil strongly enhanced soil pH.

### 3.1.5. Soil Cation Exchange Capacity

The maximum CEC concentration in S-HP soil was significantly increased to 81.61% with CB + Z + L + TB (T9) rather than the control treatment (Figure 2c). The maximum CEC content in S-LP soil was observed at 64.72% with CB + Z + L with (T8) compared with control (Figure 2c). In a previous study, Li [47] reported that natural zeolite as an amendment drastically increased the soil CEC. Mohamed [18] stated that an increase in soil CEC was observed with the addition of bamboo biochar.

### 3.1.6. Soil Dissolved Organic Carbon

The DOC proportion in S-HP soil was significantly increased 50.34% with CB + Z (T5), but CB + TB (T4) reduced the DOC proportion up to 22.7% compared to the nonamended treatment (Figure 2d). The highest increasing amount of DOC in S-LP soil was 61.0% with CB + L (T3), while DOC was reduced by 38.6% with CB (T2) compared with control (Figure 2d). In an earlier study, Liu [50] revealed enhanced DOC in soil upon the application of mixed crop straw-derived BC. Gao [51] found changes in DOC content in artificial loess, and red Cd-contaminated soils were identified after the addition of wheat straw as an amendment.

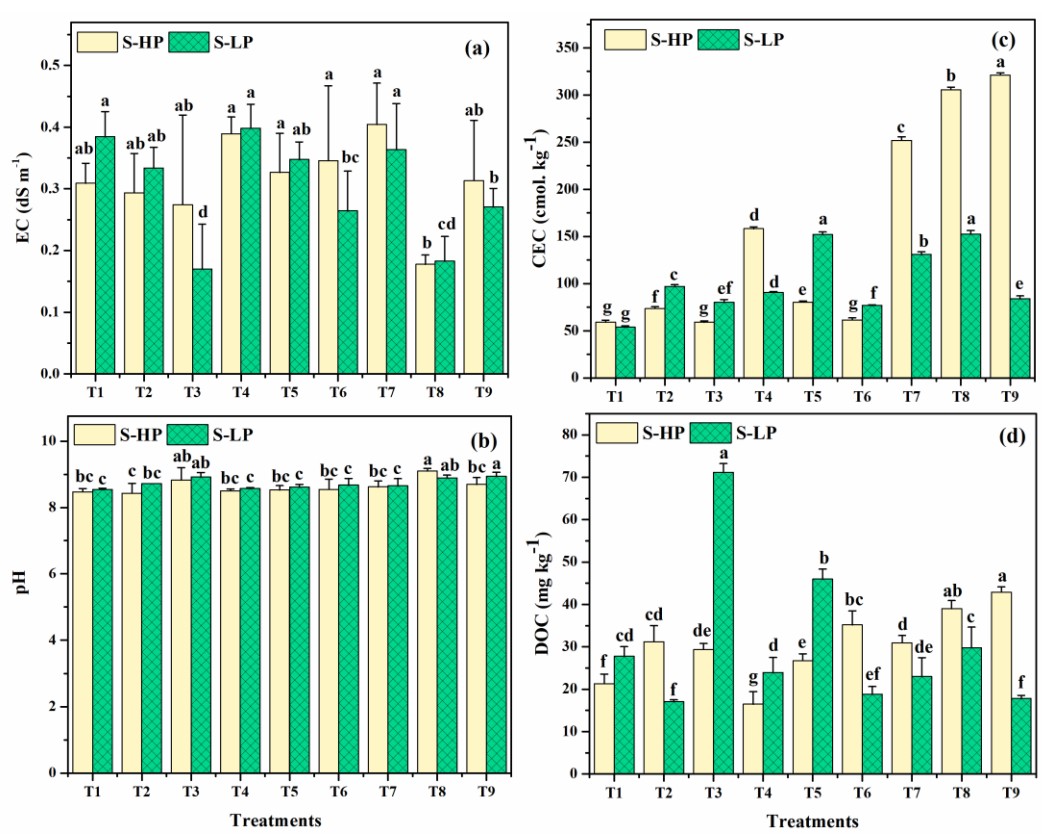

**Figure 2.** Impact of additives on soil EC after Chinese cabbage into (S-HP), and (S-LP) soils (**a**), pH (**b**), cation exchange capacity (CEC) (**c**), and dissolved organic carbon (DOC), (**d**). Values in a given column followed by the same letter are not significantly different ($p < 0.05$) using Tukey LSD test.

### 3.1.7. Soil DTPA-Extracted Cd and Zn

The maximum Cd concentration in S-HP soil was significantly reduced by 22.03% with CB + TB (T4), while CB + L + TB (T6) increased the Cd content up to 4.4% compared with nonamended soil (Figure 3a). Cd mobility in S-LP soil was nonsignificantly reduced by 29.7% with CB + L + TB (T6) than control soil (Figure 3a). Usman [52] found a stronger reduction of Cd in soil with the use of Na-bentonite and Ca-bentonite. Recently, Hamid [1] stated the Cd concentration in contaminated soil was potentially reduced with application of biochar and manure integrated with inorganic amendments.

Thus, Zhou [53] revealed reduced bioavailability of Cd and Zn in the shoots of rice with the use of limestone co-amended with zeolite as well as hydroxyhistidine with zeolite due to rising soil pH and CEC. Guo [19] reported that hydrated lime considerably immobilized the Cd in contaminated soil.

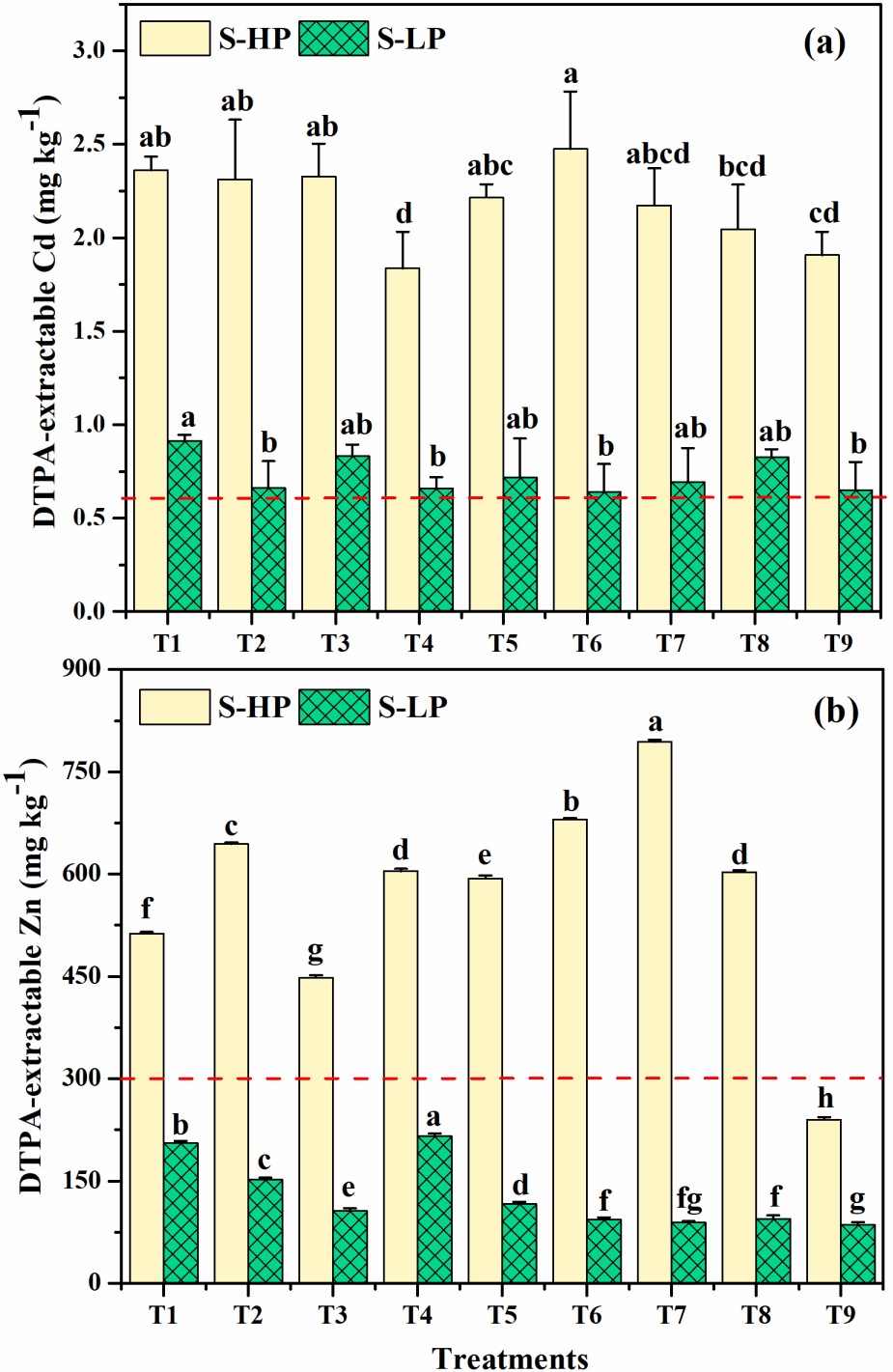

**Figure 3.** Impact of soil additives on the immobilization of DTPA-extractable Cd (**a**) and Zn (**b**) into (S-HP), and (S-LP), soils in treated samples. Values in a given column followed by the same letter are not significantly different ($p < 0.05$) using Tukey's LSD test.

Compared to the nonamended treatment, the greatest Zn mobility in S-HP soil was significantly reduced 53.2% with CB + Z + L + TB (T9), whereas CB + Z + TB (T8) enhanced Zn mobility up to 35.4%

(Figure 3b). The maximum reduction of Zn in S-LP soil obtained with CB + Z + L + TB (T9) was 58.2%, but CB + TB (T4) increased Zn mobility up to 4.9% compared with control (Figure 3b). Our findings clearly highlighted that (im) mobilization of Cd and Zn in soils might be due to variation in soil pH, OM and DOC. Similarly, Li [54] confirmed that the phytoavailability of PTE's is affected by soil chemical properties, such as EC, pH, OM and CEC. Furthermore, Cui [55] reported that apatite, lime, and charcoal as soil additives significantly immobilized Zn due to increased soil pH under long-term field conditions. Similarly, Khan and Jones [56] observed immobilization of Zn in mine-tailing sites with the addition of lime. Additionally, Zhang and Pu [57] stated that Zn and Cd were drastically fixed in urban contaminated soil with the addition of minerals. However, Abbaspour and Golchin [12] concluded that zeolite as an amendment had minimal impact on the availability and transformation of Cd and Zn in soil after a 6-month incubation period. Additionally, Almas [58] revealed that OM enhanced the mobility of Zn due to the development of organometallic groups.

### 3.1.8. Geochemical forms of Cd and Zn

Studied amendments potentially reduced the distribution of exchangeable (F1) and Fe, Mn oxide-bonded (F2), rather than organic matter-bonded (F3) and residual (F4) fractions of Cd in S-HP soil compared with control. The maximum decrease in the F1 fraction of Cd observed was 27.9% with CB + Z + L + TB (T9), while CB (T2) increased the F2 fraction of Cd up to 28.0% compared with control. Furthermore, CB (T2) reduced the F3 speciation of Cd by 25.19%, but CB + Z + L + TB (T9) potentially enhanced the Cd up to 16.7% over control. Similarly, an extreme reduction of Cd in F4 speciation (36.1%) was noted for CB + L (T3), whereas CB + L + TB (T6) maximally increased Cd by 9.34% compared to control (Figure 4a). The addition of amendments appeared to alter to the distribution of Cd in S-LP soil. Nonetheless, the greatest reductions of Cd in F1 and F2 fractions were 26.7 and 25.6% for CB + TB (T4) and CB + Z (T5), respectively, but CB + Z + TB (T7) and CB (T2) increased Cd levels to 10.9 and 14.3% compared with control. The greatest reduction in the F3 form of Cd was 8.6% with CB + Z + L + TB (T9), whereas CB + L + TB (T6) increased the Cd proportion by 12.0% compared with control. Similarly, the response of CB + TB (T4) evidently reduced the F4 fraction of Cd to a greater extent of approximately 33.3%, while CB (T2) enhanced the Cd proportion to 3.1% compared to control (Figure 4b). Regarding alkali soils, Zhao [59] found a decline in the Cd exchangeable form with the incorporation of BC at 5, 10 and 15% application rates. Moreover, Bashir [60] applied biochar and zeolite as soil amendments and found that the exchangeable form of Cd in soil was considerably reduced by 28–29.4 and 9–13%, respectively, due to increased soil pH.

The F1 form of Zn in S-HP soil was reduced by 3.6% with CB + Z + L + TB (T9), but CB (T2) increased it by 0.08% compared with nonamended soil. Compared with control, all study additives significantly enhanced the F2 form of Zn, but the maximum increase in the F2 form of Zn observed was 28.6% with CB + L + TB (T6). The maximum reduction of the F3 form of Zn was 41.9% with CB + L (T3), whereas CB (T2) increased the Zn up to 49.1% compared with control. Similarly, the greatest reduction in the F4 fraction of Zn was 42.6% with CB + L (T3), but the maximum increasing concentration of Zn observed was 29.0% with CB + TB (T4) compared with nonamended soil (Figure 4c). In S-LP soil, CB + TB (T4) was highly effective at reducing the F1 form of Zn by 38.02%, whereas CB + Z + TB (T7) potentially enhanced this form by 3.15% compared with the control treatment. The maximum reduction of the F2 form of Zn was 58.24% with CB + TB (T4), but CB + Z + L (T8) increased the Zn concentration up to 12.9% compared with control. The greatest reduction in the F3 form of Zn detected was 48.4% with CB + Z + L + TB (T9), but the highest increase in Zn concentration noted was 10.34% with CB + Z (T5) compared with control. Indeed, the F4 fraction of Zn was reduced by 10.8% with CB (T2), whereas CB + Z (T5) enhanced the Zn content up to 13.0% compared with control (Figure 4d). Castaldi [61] revealed that zeolite as an amendment significantly improved the residual form of Zn compared with control. Ahmad [62] reported that PTE's in exchangeable and reducible speciation were changed to extra steady forms using date palm leaf waste-derived BC and P-loaded BC as soil amendments. Moreover, Boguszand Oleszczuk [63] applied sewage sludge and its generated BC as soil

additives and observed that BC stimulates the alteration of accessible forms of Zn into their residual fractions over time, which may further have reduced the ecological risk associated with their existence in the atmosphere.

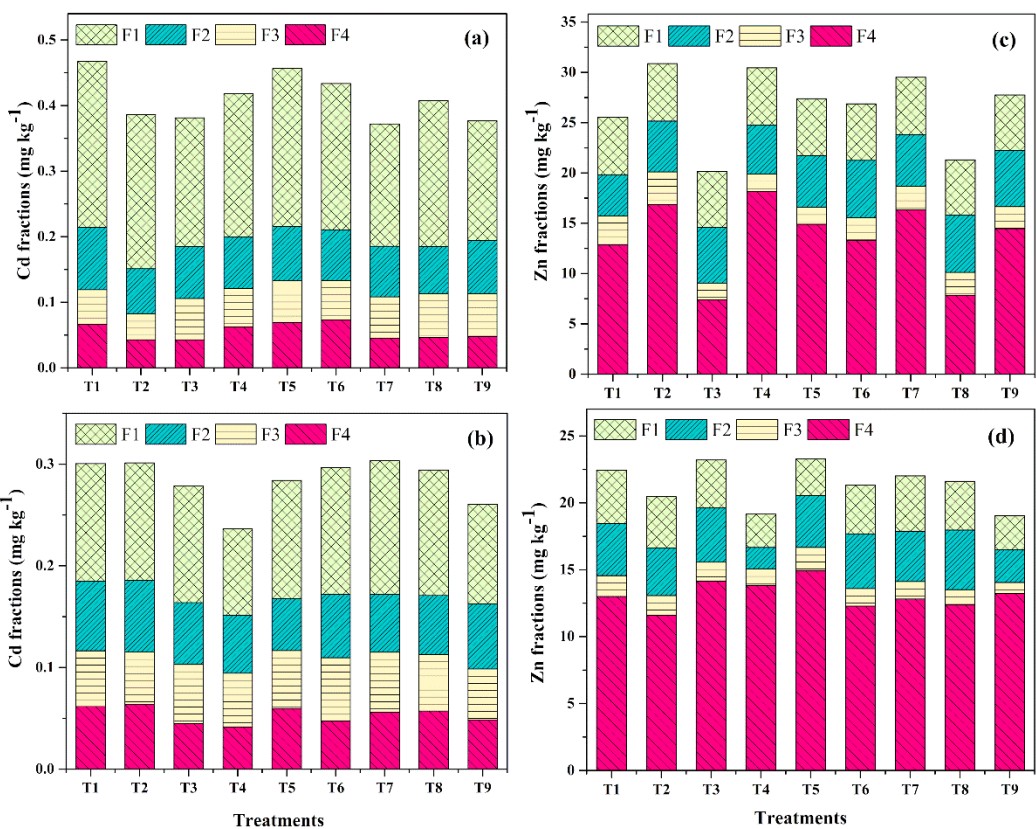

**Figure 4.** Impact of additives on distribution of PTE fractions such as F1 = exchangeable, F2 = Fe, Mg oxide-bonded, F3 = organic matter-bonded and F4 = residual of Cd in S-HP (**a**) and S-LP soils (**b**); Zn in S-HP (**c**) and S-LP soils (**d**).

### 3.1.9. Cd Uptake in Shoots and Roots

The maximum Cd uptake in the shoot of Chinese cabbage was significantly reduced by 36.0% with CB + Z + L (T8), but CB + L + TB (T6) increased the Cd uptake in shoots by 9.9% from S-HP soil compared with the control treatment (Figure 5a). Indeed, all the studied treatments drastically reduced Cd accumulation by Chinese cabbage shoot from S-LP soil compared with control. The maximum reduction of Cd in the shoot of Chinese cabbage observed was 61.35% with CB + L (T3) compared with control (Figure 5a). The Cd buildup in the root of Chinese cabbage from S-HP soil was significantly reduced by 41.8% with CB + Z + L + TB (T9), whereas CB + L + TB (T6) increased Cd accumulation in the root of Chinese cabbage up to 1.26% compared with the control treatment (Figure 5b). In case of S-LP soil, the maximum Cd reduction in the root of Chinese cabbage observed was 54.9% with CB + Z + L (T8) compared with control (Figure 5b). Consequently, Sun [64] reported that bentonite as soil additive reduced the amount of Cd in roots and shoots of (*Oryza sativa* L.) Zhang [41] revealed that rice straw-derived BC drastically reduced Cd concentration in shoots by lettuce, which was grown in lightly PTE-polluted soils. However, this treatment was not effective for immobilization and reduced the absorption by plants in heavily metal-polluted soil. More recently, Bashir [65] amended rice straw-derived BC and zeolite in acidic Cd-contaminated soil and found that Cd uptake by water spinach was significantly reduced after incorporation of amendments. However, Wen [66] assessed Cd absorption in the sorghum aerial surface with the addition of zeolite, $CaCO_3$ and $MnO_2$ as amendments.

3.1.10. Zn Uptake in Shoot and Root

Zn uptake in the shoot of Chinese cabbage from S-HP soil was significantly reduced by 9.2% with CB + L (T3), while the maximum uptake in the shoot of Chinese cabbage observed was 59.0% with CB + Z (T5) compared with control (Figure 5c). The highest reducing Zn uptake in Chinese cabbage shoots from S-LP soil observed was 58.04% with CB + Z + L + TB (T9), whereas CB + TB (T4) resulted in increased Zn uptake in Chinese cabbage shoots by 7.9% compared with control (Figure 5c). The Zn absorption in the root of Chinese cabbage from S-HP soil was potentially reduced to 29.9% with CB + Z + L + TB (T9), but CB + L + TB (T6) resulted in increased Zn absorption in Chinese cabbage roots by 52.8% compared with the control treatment (Figure 5d). The highest reduction of Zn in Chinese cabbage roots observed was 50.01% with CB + Z + L + TB (T9) from S-LP soil, while CB + TB (T4) resulted in 23.02% increased Zn absorption in Chinese cabbage roots compared with control (Figure 5d). Kumararaja [67] reported that bentonite as a soil additive reduced Zn stress in soil and uptake by amaranth. However, Wang [39] observed Zn solubility in soil and accumulation by pakchoi and Chinese cabbage after use of Ca-bentonite-pig manure co-compost.

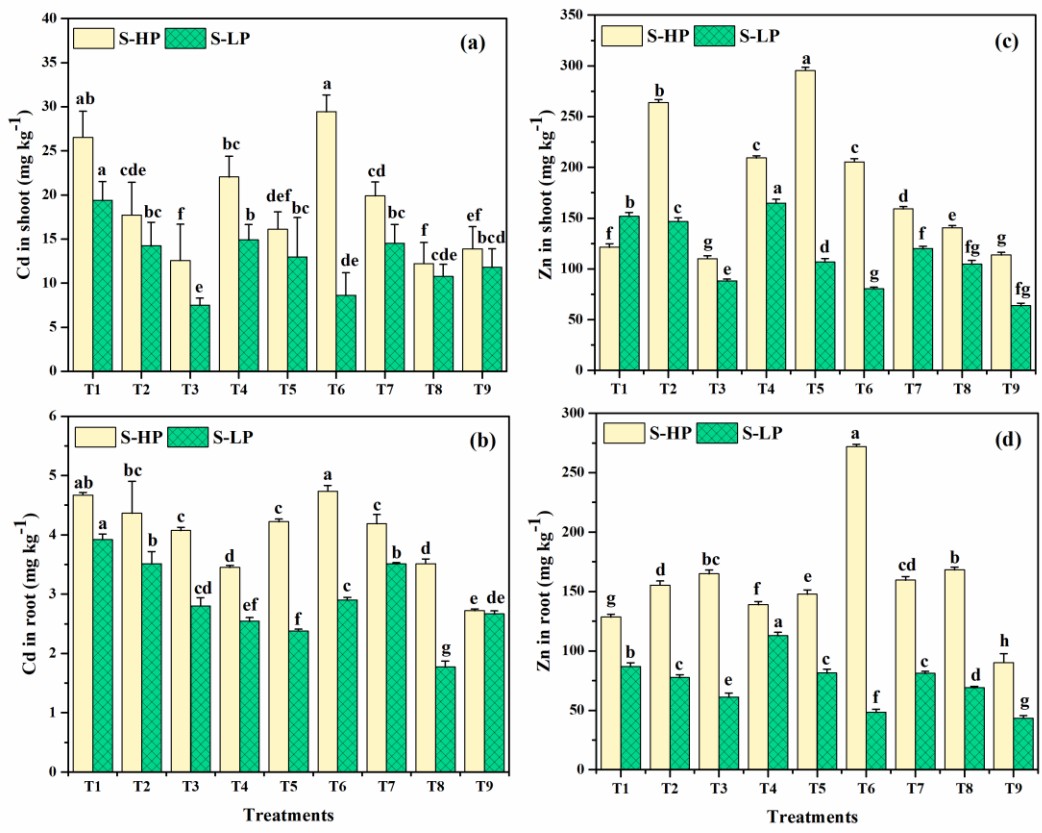

**Figure 5.** Impact of additives on the total concentration of Cd in Chinese cabbage plant shoot (**a**) and root (**b**) and Zn in Chinese cabbage plant shoot (**c**), and root (**d**) which was grown in S-HP and S-LP soils. Values in a given column followed by the same letter are not significantly different ($p < 0.05$) using Tukey's LSD test.

*3.2. Redundancy Analysis*

Redundancy analysis was performed to explore the association between DTPA-Cd and Zn in soil, plant absorption, dry biomass, chlorophyll content, EC, pH, CEC, DOC and OM in smelter heavily polluted (S-HP) and smelter low polluted (S-LP) soils (Figure 6a). Soil EC, pH, CEC, DOC and OM were plotted on the first axis (explained 22.74% of variation). However, DTPA-Cd and Zn, uptake by plants, dry biomass and chlorophyll content were plotted on the second axis (explained 29.29% of the total difference). DTPA-extractable Cd stress was significantly correlated with Cd content

in root and significantly negatively correlated with CEC. Thus, Cd content in root was negatively significantly correlated with CEC. Furthermore, shoot biomass was negatively significantly correlated with DOC in S-HP soil. Furthermore, Figure 6b shows that soil EC, pH, CEC, DOC and OM appeared on the first axis (explained 11.10% of variation). However, DTPA-Cd and Zn, uptake by plants, dry biomass and chlorophyll content appeared on the second axis (explained 47.7% of the total variation). DTPA-extractable Zn was positively associated with Cd in shoots and Zn in shoots and root biomass. Cd was positively associated with Zn in shoots and EC but negatively associated with soil pH. Zn in plant shoots was positive correlated with Zn in root and soil EC, but negatively correlated with soil pH. Zn in roots was significantly associated with EC but negatively significantly associated with soil pH. Plant shoot biomass was highly significantly correlated with EC and negatively significantly correlated with pH in S-LP soil. The analysis above demonstrates that the alteration of soil chemical parameters had significant impacts on mobilization/immobilization of PTEs in soil systems after integration of alkaline additives.

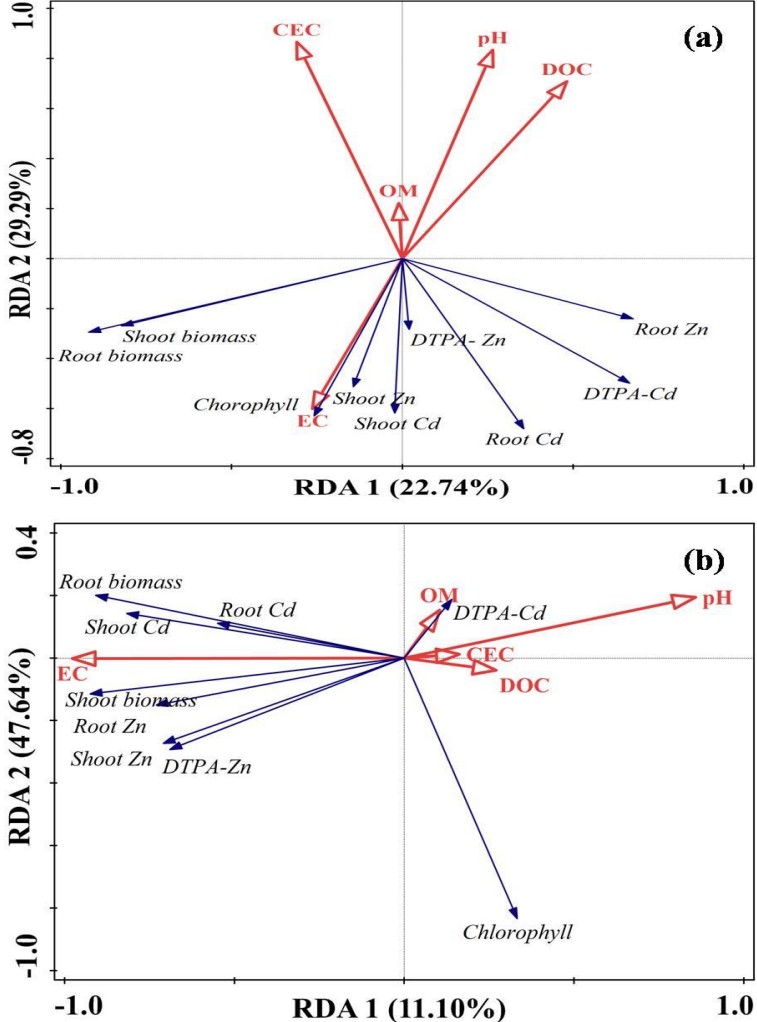

**Figure 6.** Redundancy analysis (RDA) between the DTPA-extractable Cd and Zn, uptake in plant shoot and root, dry biomass, chlorophyll content, soil EC, pH, CEC, DOC and OM from S-HP soil (**a**) and S-LP soil (**b**).

## 4. Conclusions

Overall, this study indicated that the studied amendments were highly effective for immobilization of DTPA-extractable Cd compared with Zn in S-HP and S-LP soils. Furthermore, similar behaviors of

Cd and Zn were observed in Chinese cabbage shoot and root biomass given the mobility of these TM(s) from soil systems after the addition of amendments. Interestingly, the CB + TB treatment was highly effective in promoting plant dry weight. The co-application of immobilizing agents decreased the residual fraction of Cd and increased the exchangeable fraction. In contrast, the organic matter-bonded fraction of Zn was reduced and the residual fraction increased in S-HP and S-LP soil. In general, the alteration in soil chemical characteristics could be main reason to illustrate the (im) mobilization of PTEs in treated soil samples. The collected results and findings are based on short-term pot experiments; therefore, further studies are required to verify the obtained results under field experiments with realistic and applicable rates and conditions.

**Author Contributions:** Conceptualization, A.H.L. and Z.Z.; methodology, A.H.L.; software, A.H.L. and A.C.; validation, M.M.-H. and R.I.; formal analysis, N.A.S.; investigation, Z.Z., resources, Z.Z., data curation, A.H.L, T.T.X.B. and D.D.V.; writing—original draft preparation, A.H.L. and Z.Z.; writing—review and editing, A.H.L. and E.D.; supervision, Z.Z.; project administration, Z.Z.; funding acquisition, Z.Z. All authors have carefully read and agree to the published version of the manuscript.

**Funding:** This research was financially supported by China fundamental Research Funds for the Central Universities (No. Z109021565), and Science and Technology Overall Innovation Project of Shaanxi Province in China (No. 2016KTCQ03-20) for special fund is gratefully acknowledged.

**Conflicts of Interest:** The authors declare no conflict of interest exists in the submission of this manuscript, and manuscript is approved by all authors for publication.

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
