# Peer review of "Clays, Limestone and Biochar Affect the Bioavailability and Geochemical Fractions of Cadmium and Zinc from Zn-Smelter Polluted Soils"

_sustainability, doi:10.3390/su12208606_

Round 1

Reviewer 1 Report

The introduction is extremely short. It should at least double in lenght. The research gap is not presented and the overlap with previous studies from the same authors not addressed. Much research is missing, including a review in phytoremediation and biochar and some recent studies on the topic. See:

Remediation of mining soils by combining Brassica napus growth and amendment with chars from manure waste

Effects of Manure Waste Biochars in Mining Soils

Combining phytoextraction by Brassica napus and biochar amendment for the remediation of a mining soil in Riotinto (Spain)

Use of phytoremediation and biochar to remediate heavy metal polluted soils: a review

Some work has studied the possibility of combing biochar and zeolites or lime, providing advantages which also would need to be discussed in the introduction. See:

Slow pyrolysis of biosolids in a bubbling fluidised bed reactor using biochar, activated char and lime

Thermogravimetric Analysis of biosolids pyrolysis in the presence of mineral oxides

At present there are no hypothesis and not much rationale to choose the proposed amendments.References above could assist.

References 21 and 22 are not available to the international reader. A detailed description of methods would be required here.

Line 99: This legislation is outdated. China released new legislation recently. Please, update.

A table with the physicochemical properties of both soils should be included.

Line 115: This information should be provided in this article.

There is an excessive number of abbreviations through the text. For example, avoid abbreviating trace elements.

The name of the treatments (T1 to T9) hinders following the text. Please, find something more intuitive.

There is an excessive number of decimals with percentages. Use 1 or 0 decimals.

Reference 32 should be to the original method.

Authors should do a two-way ANOVA, as it fits better their experimental design. I am afraid that the results and discussion would need a complete re-write to account for the changes in how results are presented. Please, include a table for the results of the two-way ANOVA.

Article should be proof-read by a native speaker.

Author Response

Response to reviewer’s comments

Manuscript Number: sustainability-963585
Manuscript Title: Clays, Limestone and Biochar Affect the Bioavailability and Geochemical Fractions of Cadmium and Zinc from Zn-Smelter Polluted Soils

Article Type: Research paper

Firstly, we are thankful to the reviewers for their valuable time and effort to review our manuscript and appreciate the reviewer’s comments that would certainly improve the quality of the manuscript. The responses to the comments are provided point by point as raised by the reviewers. The revisions made were highlighted in yellow color for the easy reference. Additionally, other minor corrections/revisions were made in the manuscripts that were also highlighted in yellow color. We are sure that this would satisfy reviewer’s concerns. Please find below the responses to the corrections point by point as raised by the reviewers, along with the list of changes that we have made in the revised manuscript. In addition to the changes suggested by the reviewers, we have also corrected some language and statistical errors in the manuscript.

Reviewer 1#:

Comment 1: The introduction is extremely short. It should at least double in length. The research gap is not presented and the overlap with previous studies from the same authors not addressed. Much research is missing, including a review in phytoremediation and biochar and some recent studies on the topic. See:

Remediation of mining soils by combining Brassica napus growth and amendment with chars from manure waste

Effects of Manure Waste Biochars in Mining Soils

Combining phytoextraction by Brassica napus and biochar amendment for the remediation of a mining soil in Riotinto (Spain)

Use of phytoremediation and biochar to remediate heavy metal polluted soils: a review.

Response: Thank you so much for good comments and valuable suggestions, we have paid more attention and carefully extended introduction, updated research gape, and cited above recommended papers.

Comment 2: Some work has studied the possibility of combing biochar and zeolites or lime, providing advantages which also would need to be discussed in the introduction. See:

Slow pyrolysis of biosolids in a bubbling fluidised bed reactor using biochar, activated char and lime

Thermogravimetric Analysis of biosolids pyrolysis in the presence of mineral oxides.

Response: Thank you very much for your suggestion, we have made carefully cited above two paper in introduction section and discussed the possibly of combine application of additives.

Comment 3: At present there are no hypothesis and not much rationale to choose the proposed amendments. References above could assist.

Response: Thanks for you good comment, we have carefully revise above comment as per your valuable comment.

Comment 4: References 21 and 22 are not available to the international reader. A detailed description of methods would be required here.

Response: Thanks for your nice comment, we are very sorry for that and we have carefully revise above comment as per your comment, furthermore we have described the methods of studied parameters in 2.3 section of revised manuscript.  

Comment 5: Line 99: This legislation is outdated. China released new legislation recently. Please, update.

Response: Thanks for your good comment, we have carefully revise above comment as per your valuable comment.

Comment 6: A table with the physicochemical properties of both soils should be included.

Response: Thank you so much for your good suggestion, we have carefully made a table and added in revised manuscript as per your valuable comment.

Comment 7: Line 115: This information should be provided in this article.

Response: Thank you very much, we have carefully revised above sentence as per your good suggestion.

Comment 8: There is an excessive number of abbreviations through the text. For example, avoid abbreviating trace elements.

Response: Thanks for your valuable comment. The authors have paid more attention and carefully replace the abbreviations trace elements TE(s) into potentially toxic elements PTE's.

Comment 9: The name of the treatments (T1 to T9) hinders following the text. Please, find something more intuitive.

Response: Thanks for your valuable comment. Actually, in this present work we applied calcium bentonite alone and combined with other additives, so we mentioned the treatment abbreviations to make it very clear and easily understand for readers. Please considered above treatment abbreviations in the revised manuscript.

Comment 10: There is an excessive number of decimals with percentages. Use 1 or 0 decimals.

Response: Thank you for your nice comments, we have carefully revised and reduced number of decimals in manuscript as per your valuable comment.

Comment 11: Reference 32 should be to the original method.

Response: Thank you so much for nice comments, we have carefully updated reference as per your suggestion.

Comment 12: Authors should do a two-way ANOVA, as it fits better their experimental design. I am afraid that the results and discussion would need a complete re-write to account for the changes in how results are presented. Please, include a table for the results of the two-way ANOVA.

Response: All authors are highly appreciated your nice comment. We are very sorry and apologize to do two-way ANOVA and make ANOVA table. Please consider one-way analysis of variance in the revised manuscript.  Some authors have already used combined amendment and analyzed the data in one-way analysis of variance in published papers. Please see:

 Awasthi et al., 2017. Heterogeneity of zeolite combined with biochar properties as a function of sewage sludge composting and production of nutrient-rich compost. Waste Management 68 (2017) 760–773.

Zhou et al., 2014. Effects ofcombined amendments on heavymetal accumulation in rice (Oryza sativa L.) planted on contaminated paddy soil. Ecotoxicology and Environmental Safety. 101(2014)226–232.

We are very sorry for that, please considered one-way analysis of variance in the revised paper.

Comment 13: Article should be proof-read by a native speaker.

Response: Thank you so much for your nice comment. Actually, this paper is already checked by Elsevier Web shop. The certificate is attached in the response to the comment file. Furthermore, we have carefully throughout check the grammatical and typo errors in the manuscript as per your suggestion.

Furthermore, this research paper has been carefully checked by Language Editing Services

Registered Office:

Elsevier Ltd

The Boulevard, Langford Lane,

Kidlington, OX5 1GB, UK.

Registration No. 331566771

The revised manuscript has been submitted to your reputed journal with the response of reviewer comments. We look forward to your positive response.

Thanks and best regards

Prof. Z. Q. Zhang,

College of Natural Resources and Environment,

Northwest A&F University, Yangling,               

Shaanxi Province 712100, PR China

Tel./Fax: +86 13609254113; +86 02987080055.

E-mail: [email protected] (Z. Q. Zhang).

Reviewer 2 Report

Dear authors,

The paper “sustainability-963585”  presents an interesting field of research and fits perfectly with the topics published in Sustainability. The data set is of quality and interpretations of data are good. In my opinion the manuscript is well written and acceptable after minor revision.

The topic faced by this paper is certainly of great interest and is bringing info that can contribute to the significant knowledge on the matter.

Moreover some points should revised from the authors to improve the paper.

Some remarks:

Line 47: In my opinion, it is more correct to define these elements not as toxic, but as potentially toxic elements, so I suggest using the acronym (PTE's). Zinc in particular is a fundamental element for human well-being. Obviously the acronym TE must be replaced throughout the paper.

Line 48: I would rephrase the sentence like this: “PTE’s are important inorganic pollutants in soils because they are potentially toxic for animals and humans if they enter the food chain”

Line 50/51: Reword this sentence. Written in this way, we understand that the bioavailability of all PTE's depends on the Cd

Line 178: The software is CANOCO5 not CANACO5

Line 289: fractions not factions

Line 301: Insert space between F4 and fraction

Line 302: Insert space between [58] and found

Line 304: Insert space between [59] and applied

Line 318: Insert space between Z and T5

Line 344: use always the same acronym PTE’s not HM

Line 360: use only 2 decimal places

Line 361: Insert space between [66] and reported

Author Response

Response to reviewer’s comments

Manuscript Number: sustainability-963585
Manuscript Title: Clays, Limestone and Biochar Affect the Bioavailability and Geochemical Fractions of Cadmium and Zinc from Zn-Smelter Polluted Soils

Article Type: Research paper

Firstly, we are thankful to the reviewers for their valuable time and effort to review our manuscript and appreciate the reviewer’s comments that would certainly improve the quality of the manuscript. The responses to the comments are provided point by point as raised by the reviewers. The revisions made were highlighted in yellow color for the easy reference. Additionally, other minor corrections/revisions were made in the manuscripts that were also highlighted in yellow color. We are sure that this would satisfy reviewer’s concerns. Please find below the responses to the corrections point by point as raised by the reviewers, along with the list of changes that we have made in the revised manuscript. In addition to the changes suggested by the reviewers, we have also corrected some language and statistical errors in the manuscript.

Reviewer#2

Comment 1: The topic faced by this paper is certainly of great interest and is bringing info that can contribute to the significant knowledge on the matter. Moreover, some points should have revised from the authors to improve the paper.

Response: Thank you so much. All authors are grateful for appreciate our research work.

Comment 2: Line 47: In my opinion, it is more correct to define these elements not as toxic, but as potentially toxic elements, so I suggest using the acronym (PTE's). Zinc in particular is a fundamental element for human well-being. Obviously the acronym TE must be replaced throughout the paper.

Response: Thank you so much for good comment, we have paid more attention and carefully replaced abbreviation trace elements TE(s) with potentially toxic elements PTE's as per your valuable suggestions.

Comment 3: Line 48: I would rephrase the sentence like this: “PTE’s are important inorganic pollutants in soils because they are potentially toxic for animals and humans if they enter the food chain”.

Response: Thanks for your nice comment.  We have carefully rephrased sentence in the revised manuscript as per your suggestion.

Comment 4: Line 50/51: Reword this sentence. Written in this way, we understand that the bioavailability of all PTE's depends on the Cd.

Response: Thank you so much. We have paid more attention and carefully revised above comment as per your nice comment.

Comment 5: Line 178: The software is CANOCO5 not CANACO5.

Response: Thanks, we are very sorry for that, we have carefully revised software name as per your suggestion.

Comment 6: Line 289: fractions not factions.

Response: Thank you so much for you good comment. I have carefully revised word fractions in the manuscript.

Comment 7: Line 301: Insert space between F4 and fraction.

Response: Thank you, we have carefully make space between F4 and fraction in the revised paper.

Comment 8: Line 302: Insert space between [58] and found.

Thank you again for nice comment, authors have made attention and make space b/w [58] and found.

Comment 9: Line 304: Insert space between [59] and applied.

Response: Thank you very much, we have carefully make space between [59] and applied.

Comment 10: Line 318: Insert space between Z and T5.

Response: Thank you so much, we have make space b/w Z and T5 as per nice comment.

Comment 11: Line 344: use always the same acronym PTE’s not HM.

Response: Thank you so much, we have done above comment as per your valuable comment.

Comment 12: Line 360: use only 2 decimal places.

Response: Thanks for good comment. We have carefully updated decimal in the revised manuscript as per your valuable comment.

Comment 13: Line 361: Insert space between [66] and reported.

Response: Thank you so much for valuable comments and suggestions. We have carefully done above comment as per your highly valuable comment.

The revised manuscript has been submitted to your reputed journal with the response of reviewer comments. We look forward to your positive response.

Thanks and best regards

Prof. Z. Q. Zhang,

College of Natural Resources and Environment,

Northwest A&F University, Yangling,               

Shaanxi Province 712100, PR China

Tel./Fax: +86 13609254113; +86 02987080055.

E-mail: [email protected] (Z. Q. Zhang).

Round 2

Reviewer 1 Report

In general the authors have improved their work. 

There is still some instance where too many decimals are used (see line 37 or 334 to 341 as examples)

Line 175: ANNOVA should be ANOVA

Reference 22: Should be Paz-Ferreiro and not Paz-Ferreiro1

Reference 23: The first authors should be "Álvarez, M.L."

Author Response

Response to reviewer’s comments

Manuscript Number: sustainability-963585
Manuscript Title: Clays, Limestone and Biochar Affect the Bioavailability and Geochemical Fractions of Cadmium and Zinc from Zn-Smelter Polluted Soils

Article Type: Research paper

Firstly, we are thankful to the reviewers for their valuable time and effort to review our manuscript and appreciate the reviewer’s comments that would certainly improve the quality of the manuscript. The responses to the comments are provided point by point as raised by the reviewers. The revisions made were highlighted in yellow color for the easy reference. Additionally, other minor corrections/revisions were made in the manuscripts that were also highlighted in yellow color. We are sure that this would satisfy reviewer’s concerns. Please find below the responses to the corrections point by point as raised by the reviewers, along with the list of changes that we have made in the revised manuscript. In addition to the changes suggested by the reviewers, we have also corrected some language and statistical errors in the manuscript.

Reviewer 1#:

Comment 1: There is still some instance where too many decimals are used (see line 37 or 334 to 341 as examples)

Line 175: ANNOVA should be ANOVA.

Response: Thank you so much for good comments and valuable suggestions, we have paid more attention and carefully replace the word ANNOVA with ANOVA as per your valuable suggestion.

Comment 2: Reference 22: Should be Paz-Ferreiro and not Paz-Ferreiro1.

Response: Thank you very much for your nice comments, we have carefully revised reference 22 as per your suggestion.

Comment 3: Reference 23: The first authors should be "Álvarez, M.L."

Response: Thank you so much, we paid more attention and carefully updated above reference 23 in the revised manuscript as per your valuable comment.

The revised manuscript has been submitted to your reputed journal with the response of reviewer comments. We look forward to your positive response.

Thanks and best regards

Prof. Z. Q. Zhang,

College of Natural Resources and Environment,

Northwest A&F University, Yangling,               

Shaanxi Province 712100, PR China

Tel./Fax: +86 13609254113; +86 02987080055.

E-mail: [email protected] (Z. Q. Zhang).
